# Molecular and Cytological Identification of Wheat-*Thinopyrum intermedium* Partial Amphiploid Line 92048 with Resistance to Stripe Rust and Fusarium Head Blight

**DOI:** 10.3390/plants13091198

**Published:** 2024-04-25

**Authors:** Xiaoqin Luo, Yuanjiang He, Xianli Feng, Min Huang, Kebing Huang, Xin Li, Suizhuang Yang, Yong Ren

**Affiliations:** 1Wheat Research Institute, School of Life Sciences and Engineering, Southwest University of Science and Technology, Mianyang 621010, Chinasadoneli@gmail.com (X.L.); 2Crop Characteristic Resources Creation and Utilization Key Laboratory of Sichuan Province, Mianyang Institute of Agricultural Science, Mianyang 621023, China; hy2004hy@163.com

**Keywords:** *Th. intermedium*, partial amphiploid line, nondenaturing fluorescence in situ hybridization (ND-FISH), stripe rust, fusarium head blight (FHB)

## Abstract

*Thinopyrum intermedium* (2n = 6x = 42, E^e^E^e^E^b^E^b^StSt or JJJ^s^J^s^StSt) contains a large number of genes that are highly adaptable to the environment and immune to a variety of wheat diseases, such as powdery mildew, rust, and yellow dwarf, making it an important gene source for the genetic improvement of common wheat. Currently, an important issue plaguing wheat production and breeding is the spread of pests and illnesses. Breeding disease-resistant wheat varieties using disease-resistant genes is currently the most effective measure to solve this problem. Moreover, alien resistance genes often have a stronger disease-resistant effect than the resistance genes found in common wheat. In this study, the wheat-*Th. intermedium* partial amphiploid line 92048 was developed through hybridization between *Th. intermedium* and common wheat. The chromosome structure and composition of 92048 were analyzed using ND-FISH and molecular marker analysis. The results showed that the chromosome composition of 92048 (Octoploid *Trititrigia*) was 56 = 42W + 6J + 4J^s^ + 4St. In addition, we found that 92048 was highly resistant to a mixture of stripe rust races (CYR32, CYR33, and CYR34) during the seedling stage and fusarium head blight (FHB) in the field during the adult plant stage, suggesting that the alien or wheat chromosomes in 92048 had disease-resistant gene(s) to stripe rust and FHB. There is a high probability that the gene(s) for resistance to stripe rust and FHB are from the alien chromosomes. Therefore, 92048 shows promise as a bridge material for transferring superior genes from *Th. intermedium* to common wheat and improving disease resistance in common wheat.

## 1. Introduction

Common wheat (*Triticum aestivum* L., 2n = 6x = 42, AABBDD), as the third largest cereal grain crop in terms of total global production, is the main food ration for approximately half of the global population [1,2]. Stripe rust is a prominent wheat leaf disease caused by the wheat-specialized stripe stalk rust fungus (*Puccinia striiformis* f.sp. *tritici*) and is prevalent worldwide [3,4]. Fusarium head blight/Scab (FHB) is a fungal disease of wheat spikes caused by *Fusarium graminearum* Schw. It is profoundly serious in temperate regions with humid and rainy climates [5,6]. Both stripe rust and FHB diseases can directly affect wheat yield.

For stripe rust, so far, there are 86 officially named stripe rust resistance genes, namely *Yr1*–*Yr86* [7,8]. However, the genes *Yr1*, *Yr3*, *Yr4*, *Yr9*, *Yr10*, *Yr24*, *Yr26*, and *YrCH42* have lost their resistance due to the continuous emergence of new virulence races of stripe rust. The new virulence races (CYR1, CYR8, CYR10, CYR13, CYR16, CYR18, CYR29, CYR32, CYR33, or CYR34) that emerged at the time suppressed all of the previously present stripe rust resistance genes found in the dominant cultivated wheat varieties, which ultimately led to significant yield losses [9,10,11]. While numerous effective resistance genes have now been identified, more often than not, the practical breeding of disease-resistant maincrop varieties has primarily relied on utilizing stripe rust-resistant genes that are already present in high-yielding varieties. However, their disease resistance is compromised since hybrid parents of high-yielding disease-resistant varieties have limited genetic diversity and a low presence of disease-resistance genes. As a result, their derived progeny are prone to losing disease resistance [12].

Therefore, we must investigate new stripe rust resistance genes from local varieties, wild relatives, and distantly related species [13].

As for fusarium head blight (FHB), there is a lack of effective resistance genes to deal with the epidemic, and the germplasm resources that can be used for breeding research and practical production are extremely limited [14,15]. To date, researchers have localized more than 200 different types of QTL loci for resistance to blastomycosis on wheat chromosomes, which are distributed across all wheat chromosomes [16]. However, only seven disease-resistant genes have been formally named, namely *Fhb1* to *Fhb7*. To date, *Fhb1*, *Fhb2*, *Fhb4*, and *Fhb5* have been reported to be localized on wheat chromosomes [17,18], and *Fhb3*, *Fhb6*, and *Fhb7* on the chromosomes of wheat relatives [19,20,21,22]. Among them, the *Fhb1* resistant locus is the most recognized QTL for FHB resistance, with the largest resistance effect in the world to date and stable resistance performance [23], and is most commonly used in wheat breeding. *Fhb2–Fhb7* are not widely used in practical breeding because they are inferior to *Fhb1* in terms of FHB resistance effect and stability [24].

In recent years, the occurrence of stripe rust and FHB has continued to increase in all regions due to global climate change and an increase in the number of pathogen species. Therefore, there is an urgent need to discover new and resistant sources for a wide range of wheat diseases and develop new germplasm resources. Most wheat relatives are characterized by high genetic variation and rich genetic diversity, which can effectively improve and enrich the genetic base of common wheat [25]. Among them, *Thinopyrum intermedium* (2n = 6x = 42, E^e^E^e^E^b^E^b^StSt or JJJ^S^J^S^StSt) is a valuable source of genes for common wheat genetic improvement due to its great environmental adaptability and resistance to a range of wheat diseases, including yellow dwarf, powdery mildew, and rust [26,27]. At present, many disease-resistant genes from *Th. intermedium*, including *Yr50* [28], *Lr38* [29], *Sr44* [30], *Pm40* [31], *Wsm1* [32], and *Bdv2* [33], are widely used to improve disease resistance in wheat. Wheat-*Th. intermedium* amphiploid lines, partial amphiploid lines, addition lines, substitution lines, and translocation lines containing the above genes were created by distant hybridization [34]. They are particularly important intermediate materials in the selection process for wheat varieties [35].

The disease resistance of these wheat-*Th. intermedium* lines was significantly important due to the presence of certain chromosomes or chromosome segments from *Th. intermedium*. The research and utilization of genes from *Th. intermedium* have played a positive role in promoting wheat germplasm innovation and genetic improvement [36,37,38]. To cultivate new germplasm resources, we aim to investigate and evaluate the chromosome composition, agronomic traits, stripe rust, and FHB resistance of 92048 in this study. It aims to provide a theoretical basis for fully utilizing the breeding potential of *Th. intermedium* in the future to broaden the genetic background and improve the disease resistance of common wheat.

## 2. Results

### 2.1. ND-FISH Analysis of 92048 by Multiple Oligo Probes

The sequential ND-FISH was used to analyze the chromosome composition and structure of 92048 using Oligo-pTa535-1, Oligo-pSc119.2-1, Oligo-B11, and Oligo-pDb12H. Oligo-pTa535-1 and Oligo-pSc119.2-1 could recognize all 42 chromosomes of common wheat 1A–7D. Oligo-B11 and Oligo-pDb12H enabled us to identify the J^s^, J, and St genomes of *Th. intermedium*. Because Oligo-B11 can produce abundant signals on the entire chromosome of the St-chromosomes, telomeres and subtelomeres of the J-chromosomes, and telomeres, subtelomeres, and centromeres of the Js-chromosomes. Whereas Oligo-pDb12H will only produce signals on the double arms of the J^s^-chromosomes. Thus, the probes Oligo-B11 and Oligo-pDb12H can be used for ND-FISH to clearly distinguish the *Th. intermedium* chromosomes into three groups (St, J and J^s^) with 14 chromosomes each [39].

The results of the sequential ND-FISH analysis showed that the root tip meristem cells of 92048 contained a total of 56 chromosomes, consisting of 42 wheat chromosomes arranged in seven pairs of *Th. intermedium* chromosomes (Figure 1a,b).

The seven pairs of *Th. intermedium* chromosomes in 92048 contained strong Oligo-B11 signals; five pairs contained Oligo-pTa535-1 signals; and one pair contained weak signals of Oligo-pSc119.2-1 (Figure 1). According to the signal patterns of the four probes, it can be determined that line 92048 contained 1J, 5J-St, 7J, 2J^s^, 5J^s^-St, 4St, and 5St chromosomes. The FISH signals of probes Oligo-pDb12H, Oligo-pTa535-1, and Oligo-pSc119.2-1 confirm the presence of 5J-St and 2J^s^ chromosomes (Figure 2). The FISH signal patterns of the seven pairs of *Th. intermedium* chromosomes in 92048 are consistent with the results reported by Yu et al. [39,40]. However, there may also be small segments of chromosomal translocations and infiltrations in 92048 that cannot be detected precisely by ND-FISH experiments and must be confirmed using molecular labeling techniques to further verify its chromosomal composition.

### 2.2. Molecular Marker Analysis

Liu et al. [41] used chromosome-specific STS markers of *Th. Intermedium* to select five random STS markers for each of the J, J^s^, and St chromosomes belonging to different homologous groups. This resulted in a total of 105 STS markers, which were then used for PCR amplification of the genomic DNAs of 92048, *Th. intermedium*, Longchun 10, and Gan 630 (Appendix A). The results showed that 83 of the 105 primer pairs amplified the target band in *Th. intermedium* (Appendix A), and no target band was amplified in Longchun 10 and Gan 630. Four pairs of 1J-specific STS primers (Figure 3a), two pairs of 2J^s^-specific STS primers (Figure 3d), three pairs of 4St-specific STS primers (Figure 3f), two pairs of 5J-specific STS primers (Figure 3b), three pairs of 5St-specific STS primers (Figure 3g), two pairs of 5J^s^-specific STS primers (Figure 3e), and four pairs of 7J-specific STS primers (Figure 3c) amplified the target product from 92048 and *Th. intermedium*. Therefore, the seven pairs of *Th. intermedium* chromosomes in 92048 were identified as 1J, 5J, 7J, 2J^s^, 5J^s^, 4St, and 5St. In addition, 10–20% of the STS markers corresponding to chromosomes 1St, 2J, 3J^s^, 3St, 4J, 4J^s^, and 6J^s^ exhibited amplified target bands in 92048 (Appendix A). These chromosomes are likely to be small fragments of *Th. intermedium* chromosomes present in translocated or infiltrated 92048.

Finally, combining the results of the above ND-FISH experiments, it is known that 92048 is a wheat-*Th. intermedium* partial double diploid line (octoploid *Trititrigia*) containing 42 wheat chromosomes and seven pairs of *Th. intermedium* chromosomes.

### 2.3. Identification of Resistance to Stripe Rust and FHB

To evaluate the resistance response of 92048 to stripe rust and FHB, 92048, Longchun 10, and Gan 630 plants were inoculated with stripe rust and FHB in the greenhouse and field (Figure 4 and Figure 5; Table 1 and Table 2). Infection types (ITs) and degree of severity (DS) of stripe rust were assessed according to the nine-level reaction type method, with the DS of stripe rust being the percentage of leaf area occupied by the uredospore: 0–100 (%) [42]. The ITs and DS of FHB were assessed according to the four-level reaction type method, with the DS of FHB being the percentage of the number of infected spikelets in a single spikelet: 0–100 (%) [43].

During the seedling stage, 92048 was highly resistant (HR) to a mixture of stripe rust races: CYR32, CYR33, and CYR34 (Figure 4a; IT = 1, DS = 5%); Longchun 10 and Gan 630 were highly susceptible (HS) to a mixture of stripe rust races: CYR32, CYR33, and CYR34 (Figure 4b,c; IT = 8, DS = 90%), suggesting that the alien chromosomes in 92048 had seedling resistance gene(s) to stripe rust.

During the adult plant stage, 92048 was highly resistant (HR) to stripe rust (Figure 5a; IT = 1, DS = 10%) and FHB (Figure 5d; IT = 1, DS = 5.88%); Longchun 10 was highly susceptible (HS) to stripe rust (Figure 5b; IT = 8, DS = 90%) and moderately susceptible (MS) to FHB (Figure 5e; IT = 3, DS = 56.25%); and Gan 630 was highly susceptible (HS) to stripe rust (Figure 5c; IT = 8, DS = 90%) and FHB (Figure 5f; IT = 4, DS = 100%). This result indicates that the alien or wheat chromosomes in 92048 had adult plant resistance gene(s) to stripe rust and FHB. In addition, since the resistance to stripe rust and FHB of 92048 was significantly higher than that of 92048’s parents, Longchun 10 and Gan 630, it is highly likely that the stripe rust and FHB resistance in 92048 was derived from the alien chromosomes.

### 2.4. Assessment of Agronomic Performance

The seeds of 92048 were elongated and not full enough (Figure 6a,b). The spikes of 92048 had no awns, and the glumes of the spikes were thicker (Figure 6c,d). Comparing 92048 with its parents, Longchun 10 and Gan 630, the agronomic performance of 92048 was unsatisfactory. The agronomic traits survey results of 92048 showed that plant height, thousand kernel weight, and grain width were significantly lower than those of 92048’s parents, Longchun 10 and Gan 630; tiller number per plant, spike length, total spikelet number per spike, and grain length of 92048 did not differ from those of Longchun 10; however, tiller number per plant, spike length, and grain length were significantly higher than those of Gan 630 (Figure 6e). Based on the above results, there is a strong probability that alien chromosomes in 92048 are the source of the poor agronomic performance of 92048. Therefore, 92048 is an intermediate material for wheat breeding, as it can further crossbreed with wheat possessing excellent agronomic traits. This will facilitate the development of new wheat varieties that exhibit outstanding agronomic traits and disease resistance.

## 3. Discussion

Two main issues plaguing wheat production and breeding are the spread of pests and illnesses and the inability of current wheat varieties to meet the changing demands of humankind [2].

Wheat stripe rust is a significant disease that is widespread, extremely destructive, and influential in wheat production [4]. Although a large number of effective resistance genes have now been identified and applied to wheat breeding, they are still insufficient to cope with the evolving virulence races. Often, after a new batch of wheat disease-resistant varieties has been selected and widely disseminated, new virulence races that are compatible with them subsequently appear, bringing serious challenges to wheat production and wheat disease resistance breeding [44,45,46]. FHB is known as the “cancer” of wheat and is also common in wheat-producing countries worldwide. It is particularly severe in temperate regions with humid and rainy climates [5,15]. Unlike stripe rust, FHB lacks a large number of effective resistance genes to cope with the epidemic, and among the few resistance genes that have been reported, only *Fhb1* is stable and has been put into practical production [16,20].

Using alien genes is a key strategy and an effective way to develop better wheat cultivars, which is important for broadening the genetic background of wheat and sustaining control of wheat disease [10]. *Th. intermedium* has excellent stress and disease resistance, such as cold and drought tolerance, high resistance to FHB, and immunity to powdery mildew and rust. At the same time, *Th. intermedium* has a high hybrid affinity with wheat; it is an excellent distant relative in the genetic improvement of wheat [47]. The stripe rust resistance gene *Yr50* has been localized in *Th. intermedium* [28] and introduced into common wheat to cultivate disease-resistant wheat varieties. There is no report on the localization of FHB resistance genes in *Th. intermedium*. However, the *Fhb7* gene has localized on the long arm of chromosome 7E in *Elytrigia elongata* [22,48]. *Th. intermedium* and *Elytrigia elongata* have an identical chromosome set (E (=J) [49,50]), so it is highly likely that *Th. intermedium* also possesses the FHB resistance gene(s) on the 7E chromosome.

In this study, we discovered that the wheat-*Th. intermedium* partial diploid line is highly resistant (HR) to a mixture of stripe rust races (CYR32, CYR33, and CYR34) during the seedling stage and highly resistant (HR) to stripe rust and FHB in the field during the adult plant stage. This suggests that there are likely gene(s) present on either the alien or wheat chromosomes in 92048 that enhance resistance at all stages (ARS) to CYR32, CYR33, and CYR34 and adult plant resistance (APR) to FHB. ARS can be easily overcome by virulence races, but it is highly resistant to disease and can be quickly bred into resistant varieties, which is highly effective in controlling disease epidemics. While APR is relatively weak, it has a wide range of resistance, is more durable, and is not easily overcome by new virulence races [51].

Currently, foreign breeders are advocating the use of more APR genes and less ASR genes [52]. However, when the characteristics of disease transmission and breeding strategies change, relying only on genes for APR may not be the best method to cope with disease susceptibility throughout the reproductive period of wheat. If new wheat varieties are bred by aggregating genes for ASR and APR genes together, the new wheat varieties will maintain adequate disease resistance throughout the reproductive period, which will reduce the damage of wheat diseases [12]. CYR32, CYR33, and CYR34 are now the dominant virulence races of stripe rust in China, which are extremely harmful to wheat production. Therefore, in the future, 92048 could be used as a new gene polymerization material against CYR32, CYR33, and CYR34 to enhance stripe rust resistance in new wheat varieties.

In addition, multi-resistance breeding is also a research hotspot in wheat breeding research. Nowadays, the weather is complex and variable, various wheat diseases occur more frequently, and multiple wheat diseases are prevalent in the same region [53]. Breeding wheat varieties with multiple resistances is an effective way to solve this problem [54]. In the study of Li et al. [55], 2J^s^ and 4J^s^ chromosomes were found to enhance stripe rust resistance in wheat during the adult plant stage. Long Dan et al. found that hybrid-derived progeny containing the E (=J) chromosomes had high resistance to stripe rust [35]. In this study, we found that 92048 carried 1J, 5J-St, 7J, 2J^s^, 5J^s^-J, 4St, and 5St chromosomes and showed high resistance to stripe rust and FHB. It is speculated that the 1J, 5J, 7J, and 2J^s^ chromosomes may be the source of 92048’s stripe rust and FHB resistance, while the 7J chromosome may contain both stripe rust and FHB resistance genes. Therefore, the 7J chromosome may be used in the future as a new germplasm resource for breeding multiple resistances in wheat and to further reduce the time and cost of breeding disease resistance in wheat. In the future, we need to conduct a detailed study on the identity information of the stripe rust and FHB resistance gene(s) in 92048.

## 4. Materials and Methods

### 4.1. Plant Materials

*Th. intermedium* Z1141 (2n = 6x = 42, E^e^E^e^E^b^E^b^StSt or JJJ^s^J^s^StSt) exhibits high levels of disease resistance to FHB and immunity to stripe rust. Gan 630 and Longchun 10, two cultivated wheat varieties in Gansu Province, are highly susceptible to stripe rust and FHB. *Th. intermedium* was used as the parent, successively crossed with Gan 630 and Longchun 10, and then self-pollinated for nine generations. The hybridization method is (Gan 630/*Th. intermedium*) F_1_/(Longchun 10/Gan 630) F_1_. A stable wheat-*Th. intermedium*-derived line bred by Ni Jianfu of the Gansu Academy of Agricultural Sciences was created: 92048.

### 4.2. Molecular Cytogenetic Analyses

The chromosome composition and structure of 92048 were analyzed by nondenaturing fluorescence in situ hybridization (ND-FISH). The preparation of root tip cells was carried out using the method created by Han et al. [56]. Root tips of wheat seedlings 1.5 to 2.0 cm long were taken and placed in 1 MPa of N_2_O for 2 h, then fixed in 90% acetic acid for 15 min. The root tips were washed 3 times with ddH_2_O, and the creamy white parts of the root tips were cut and transferred into an enzymatic solution (1% pectinase + 2% cellulase) and water bath at 37 °C for 1h. The root tips were washed three times with 70% ethanol, and then the root tip tissues were ground to make a cell suspension of the apical meristematic tissue zone. Finally, the slides were titrated and observed for chromosome split-phase status, and the slides with a better split-phase were selected for subsequent ND-FISH. ND-FISH was performed using synthetic oligonucleotide probes: Oligo-pSc119.2-1 [57], Oligo-pTa535-1 [57], Oligo-B11 [58], and Oligo-pDb12H [39] (Table 3). Each chromosome was distinguished using the FISH karyotype described by Tang et al. [57], Yu et al. [39,40], and Cui et al. [44]. The ND-FISH process was carried out according to the methods described by Xi et al. [58]. An epifluorescence microscope (BX51, Olympus Corporation, Tokyo, Japan) equipped with a cooled charge-coupled device camera operated with HCIMAGE Live software (Hamamatsu Corporation, Sewickley, PA, USA) was used to take images.

### 4.3. Molecular Marker Analysis

In order to determine the chromosome composition of 92048, the leaf DNA of 92048 was examined using chromosome-specific STS molecular markers (Appendix A) of *Th. intermedium* developed and validated by Liu et al. [41]. DNA was extracted from the young leaves of *Th. intermedium*, 92048, Longchun 10, and Gan 630 by CTAB. The PCR process was carried out according to the methods described by Liu et al. [41]. Finally, the PCR-amplified products were electrophoresed on an 8% PAGE gel following the method of Cui et al. [59].

### 4.4. Stripe Rust and FHB Resistance Tests

To evaluate the stripe rust and FHB resistance of 92048 and its parents, Longchun 10 and Gan 630, under greenhouse settings, we used a mixture of stripe rust (*P. striiformis* f. sp. *tritici*, Pst) races (CYR32, CYR33, and CYR34) and one FHB race (F_15_). At the same time, in order to characterize resistance to both stripe rust and FHB in the adult stage, we simultaneously sowed 92048 and its parents, Longchun 10 and Gan 630, in a field in Mianyang, China, where stripe rust can develop naturally. We artificially inoculated the middle of the wheat spike with pathogenic spores of F_15_ by single-flower drop injection. All materials were field tested at the Science and Technology Park of Southwest University of Science and Technology in Mianyang, China, from 2021 to 2023. Stripe rust infection types (ITs) and disease severity (DS) were assessed after the onset of stripe rust in the field. The type and severity of the FHB infection were evaluated within 21 to 30 days after erythromycin inoculation. Stripe rust ITs were assessed using the nine-level reaction type method [42]. FHB ITs and disease severity (DS) were assessed using the four-level reaction type method [43].

## 5. Conclusions

The identification of the wheat-*Th. intermedium* partial amphiploid line 92048 revealed that it carried seven pairs of *Th. intermedium* chromosomes (1J, 5J-St, 7J, 2J^s^, 5J^s^-J, 4St, and 5St). This was determined using ND-FISH and molecular marker analysis. Additionally, recombination and translocation events were observed between chromosomes 5J and 5J^s^ and the St chromosomes. Moreover, we identified that 92048 is highly resistant (HR) to stripe rust and FHB. However, the agronomic traits of 92048 do not perform well. In particular, the thousand kernel weight of 92048 was significantly lower than that of Longchun 10 and Gan 30, the parents of 92048. The wheat-*Th. intermedium* line is a great potential resource for resistance wheat breeding; however, further research is needed to identify the specific genes responsible for resistance in 92048, which will help utilize this line more effectively in wheat breeding programs.

## Figures and Tables

**Figure 1 plants-13-01198-f001:**
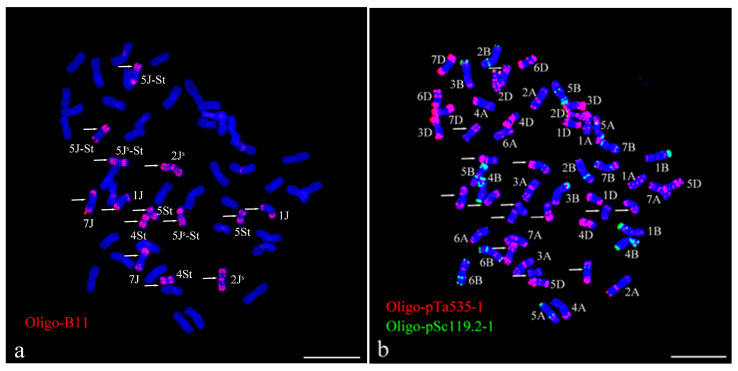
Non-denaturing fluorescence in situ hybridization (ND-FISH) identification of 92048. Sequential ND-FISH analyses of 92048 using Oligo-pTa535-1 (red), Oligo-pSc119.2-1 (green), and Oligo-B11 (red) as probes on root tip metaphase chromosomes of 92048 (**a**–**c**). The white arrows indicate *Th. intermedium* chromosomes (**a**,**b**), and the white triangles indicate translocation chromosomes (**c**). Chromosomes were counterstained with DAPI (blue). Scale bar: 10 µm.

**Figure 2 plants-13-01198-f002:**
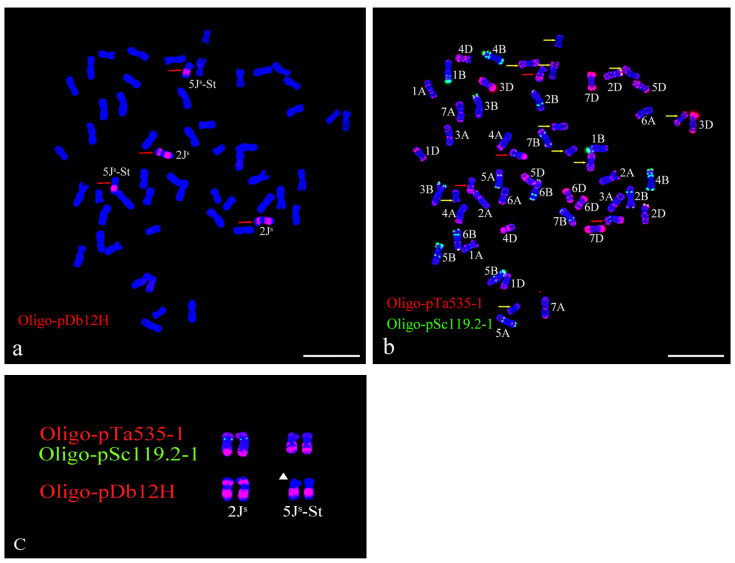
Non-denaturing fluorescence in situ hybridization (ND-FISH) identification of 92048. Sequential ND-FISH analyses of 92048 using Oligo-pTa535-1 (red), Oligo-pSc119.2-1 (green), and Oligo-pDb12H (red) as probes on root tip metaphase chromosomes of 92048 (**a**–**c**). The red arrows indicate J^s^ chromosomes (**a**,**b**), the yellow arrows indicate J and St chromosomes, and the white triangles indicate translocation chromosomes (**c**). Chromosomes were counterstained with DAPI (blue). Scale bar: 10 µm.

**Figure 3 plants-13-01198-f003:**
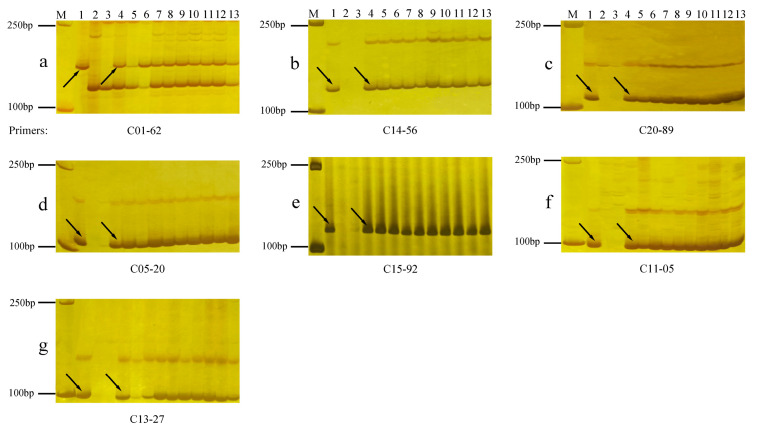
Polyacrylamide gel electrophoresis analysis of the amplification results of STS-specific markers for the *Th. intermedium* chromosomes in two cultivated wheat varieties and ten 92048 plants. M: DNA ladder; 1: *Th. intermedium* plants; 2: Gan 630; 3: Longchun 10; 4–13: 92048 plants. The black arrows indicate polymorphic bands. (**a**): 1J, (**b**): 5J, (**c**): 7J, (**d**): 2J^s^, (**e**): 5J^s^, (**f**): 4St, and (**g**): 5St.

**Figure 4 plants-13-01198-f004:**
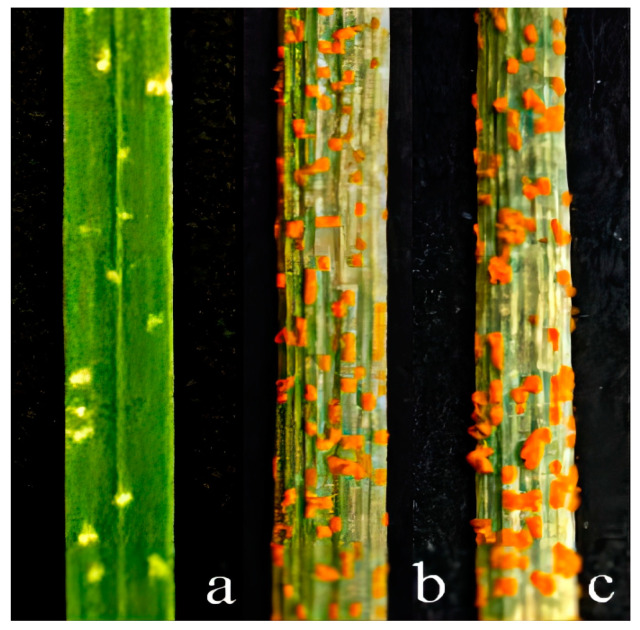
Leaf resistance to stripe rust in 92048, Longchun 10, and Gan 630 plants (the seedling stage) (**a**–**c**).

**Figure 5 plants-13-01198-f005:**
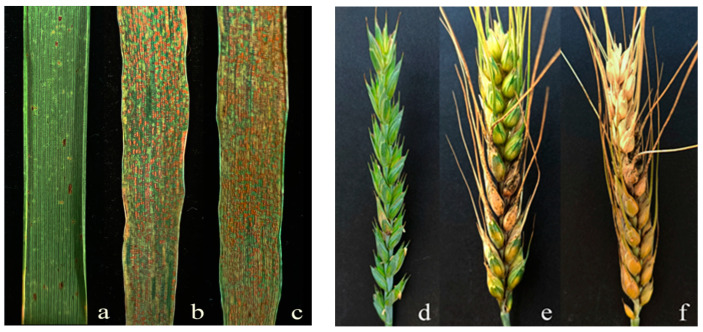
Leaf resistance to FHB in 92048, Longchun 10, and Gan 630 (**a**–**c**); Spike resistance to stripe rust in 92048, Longchun 10, and Gan 630 (**d**–**f**) (the adult plant stage).

**Figure 6 plants-13-01198-f006:**
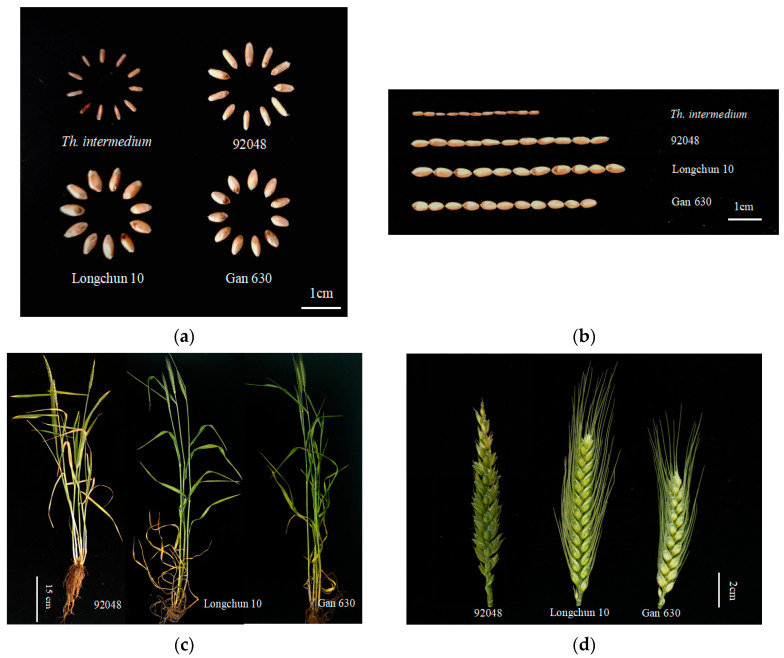
Agronomic performance of 92048 and its parents, Longchun 10 and Gan 630. Seed characteristics (**a**,**b**), plants (**c**), spike characteristics (**d**), and agronomic evaluation (**e**) of 92048 and its parents, Longchun 10 and Gan 630. Values followed by the same letters in the same column were not significantly different at the 0.05 probability level according to the LSD test. Error bars represent the standard deviation of the mean.

**Table 1 plants-13-01198-t001:** The tested materials resistance to stripe rust (the seedling stage).

Material Name	Infection Types (ITs)	Degree of Severity (DS)	Judged Resistances or Susceptibilities
92048	1	5%	HR
Longchun 10	8	90%	HS
Gan 630	8	90%	HS

**Table 2 plants-13-01198-t002:** The tested materials resistance to stripe rust and FHB (the adult plant stage).

	Stripe Rust			FHB		
Material Name	Infection Types (ITs)	Degree of Severity (DS)	Judged Resistances or Susceptibilities	Infection Types (ITs)	Degree of Severity (DS)	Judged Resistances or Susceptibilities
92048	1	5%	HR	1	5.88%	HR
Longchun 10	8	90%	HS	3	56.25%	MS
Gan 630	8	90%	HS	4	100%	HS

**Table 3 plants-13-01198-t003:** Sequences of the oligonucleotide probes used in this study.

Probe	Oligonucleotide Sequence	5′-Modification	References
Oligo-pSc119.2-1	5′CCGTTTTGTGGACTATTACTCACCGCTT TGG GGTCCCATAGCTAT 3′	5′-Cy5	[57]
Oligo-pTa535-1	5′AAAAACTTGACGCACGTCACGTACAAATTGGACAAACTCTTTCGGAGTATCAGGGTTTC 3′	5′-Tamra	[57]
Oligo-B11	5′TCCGCTCACCTTGATGACAACATCAGG TGGAATTCCGTTCGAGGG 3′	5′-FAM	[58]
Oligo-pDb12H	5′TCAGAATTTTTAGGATAGCAGAAGTAT TCG AAATACCCAGATTGCTACAG 3′	5′-FAM	[39]

## Data Availability

Data are contained within this study.

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
