# Peer review of "Molecular and Cytological Identification of Wheat-Thinopyrum intermedium Partial Amphiploid Line 92048 with Resistance to Stripe Rust and Fusarium Head Blight"

_plants, 2024, doi:10.3390/plants13091198_

Round 1
Reviewer 1 Report
Comments and Suggestions for Authors
This manuscript reported a wheat-Thinopyrum Intermedium partial amphiploid line 92048 with resistance to stripe rust and fusarium head blight. This line may be used in wheat breeding and enrich resistance genes. My minor concerns are as follows:
The ND-FISH should be explained when it appeared at the first time.
“Thinopyrum Intermedium” should be “Thinopyrum intermedium”. After first time use, it should be “Th. intermedium”.
L24 “..... highly resistant…” should be “…highly resistant to…”
L41 “So far” should be “so far”
L79 “,et al.” should be “, et al.”
L82 there more than one space before “that”?
L89 “[36-38” should be “[…]”
L89-96 the sentence is too long. It should be rephrased.
L101 “Oligo-pTa535-1, Oligo-pSc119.2-1” should be “Oligo-pTa535-1 and Oligo-pSc119.2-1”
L104 “chromosome number 92048…” should be “The chromosome number of 92048…”
L164, IT and DS should be explained when they appeared at the first time.
L191-192, the sentence was interrupted and should be revised.
L214 more space before “the”?
L217 Should be “resistant to …”
L255 “FHB..” should be “FHB.”
L258 developed by?
L268, It seemed the title of Table 3 is wrong?
L277 Under should be under?
L294-295 the sentence should be revised.
Author Response
Dear reviewer,
Our feedback on all your comments is in the attached Word document. Please review it.

Reviewer 2 Report
Comments and Suggestions for Authors
The figures are poor and fuzzy. Authors also must show karyotype of wheat lines- complete cell and then the karyotype. It is difficult to follow the numbering of chromosomes on a complete cell.
Similarly combined GISH/FISH evidence is needed to show the different genomic origin of Th intermedium chromosomes.
Comments on the Quality of English Language
English is poor and thus the ms. cannot be published as is. Just two examples:
on line 56:
As for FHB, there is a lack of a large number of effective resistance genes to deal with
the epidemic of blast disease,
FHB is not same as blast disease.
Lines 103-104:
It is saying "chromosome number 92048 was 56 (" which means there is a chromosome called 92048 which was 56??
But the correct sentence will be "Chromosome number of line 92048 was 56"
And this type of errors and flaws in English language are everywhere.
I cannot recommend publication because of poor English.
Author Response
First of all, I would like to thank you very much for your comments on my manuscript, which have been very helpful in revising the manuscript. My feedback on your comments is as follows:
1、Reviewer’s comment: on line 56:
As for FHB, there is a lack of a large number of effective resistance genes to deal with the epidemic of blast disease,FHB is not same as blast disease.
Author's response:Line 56 “blast disease” was changed to “FHB”
2、Reviewer’s comment: Lines 103-104:
It is saying "chromosome number 92048 was 56 (" which means there is a chromosome called 92048 which was 56??
But the correct sentence will be "Chromosome number of line 92048 was 56"
Author's response: Lines 103-104 "chromosome number 92048 was 56” was changed to “92048 contained a total of 56 chromosomes”
3、Reviewer’s comment: I cannot recommend publication because of poor English.
Author's response: I have used MDPI's English editing service to optimize the English within the manuscript.
Finally, we would like to thank you again for your attention and support, and hope that you will continue to follow our research work and offer more valuable comments and suggestions.
Reviewer 3 Report
Comments and Suggestions for Authors
Dear Authors,
All my comments on the manuscript are in the attached Word document.

Author Response

(The authors gave the same response as above.)

Round 2
Reviewer 2 Report
Comments and Suggestions for Authors
The cytogenetic analysis is sound and I recommend the paper for publication.
Author Response
Dear reviewer,
I would like to thank you very much for your comments on my manuscript, which have been very helpful in revising the manuscript. I accept all your comments and have revised the manuscript.